# Learning to Linearize Under Uncertainty

**Ross Goroshin**[*1] **Michael Mathieu**[*1] **Yann LeCun**[1,2]
[1]Dept. of Computer Science, Courant Institute of Mathematical Science, New York, NY
[2]Facebook AI Research, New York, NY
{goroshin,mathieu,yann}@cs.nyu.edu

## Abstract

Training deep feature hierarchies to solve supervised learning tasks has achieved state of the art performance on many problems in computer vision. However, a principled way in which to train such hierarchies in the unsupervised setting has remained elusive. In this work we suggest a new architecture and loss for training deep feature hierarchies that *linearize* the transformations observed in unlabeled natural video sequences. This is done by training a generative model to predict video frames. We also address the problem of inherent *uncertainty* in prediction by introducing latent variables that are non-deterministic functions of the input into the network architecture.

## 1    Introduction

The recent success of deep feature learning in the supervised setting has inspired renewed interest in feature learning in weakly supervised and unsupervised settings. Recent findings in computer vision problems have shown that the representations learned for one task can be readily transferred to others [10], which naturally leads to the question: does there exist a generically useful feature representation, and if so what principles can be exploited to learn it?

Recently there has been a flurry of work on learning features from video using varying degrees of supervision [14][12][13]. Temporal coherence in video can be considered as a form of weak supervision that can be exploited for feature learning. More precisely, if we assume that data occupies some low dimensional "manifold" in a high dimensional space, then videos can be considered as one-dimensional trajectories on this manifold parametrized by time. Many unsupervised learning algorithms can be viewed as various parameterizations (implicit or explicit) of the data manifold [1]. For instance, sparse coding implicitly assumes a locally linear model of the data manifold [9]. In this work, we assume that deep convolutional networks are good parametric models for natural data. Parameterizations of the data manifold can be learned by training these networks to *linearize* short temporal trajectories, thereby implicitly learning a local parametrization.

In this work we cast the linearization objective as a frame *prediction* problem. As in many other unsupervised learning schemes, this necessitates a generative model. Several recent works have also trained deep networks for the task of frame prediction [12][14][13]. However, unlike other works that focus on prediction as a final objective, in this work prediction is regarded as a proxy for learning representations. We introduce a loss and architecture that addresses two main problems in frame prediction: (1) minimizing $L^2$ error between the predicted and actual frame leads to unrealistically blurry predictions, which potentially compromises the learned representation, and (2) copying the most recent frame to the input seems to be a hard-to-escape trap of the objective function, which results in the network learning little more than the identity function. We argue that the source of blur partially stems from the inherent unpredictability of natural data; in cases where multiple valid predictions are plausible, a deterministic network will learn to average between all the plausible predictions. To address the first problem we introduce a set of latent variables that are non-deterministic

---

functions of the input, which are used to explain the unpredictable aspects of natural videos. The second problem is addressed by introducing an architecture that explicitly formulates the prediction in the linearized feature space.

The paper is organized as follows. Section 2 reviews relevant prior work. Section 3 introduces the basic architecture used for learning linearized representations. Subsection 3.1 introduces "phase-pooling"–an operator that facilitates linearization by inducing a topology on the feature space. Subsection 3.2 introduces a latent variable formulation as a means of learning to linearize under uncertainty. Section 4 presents experimental results on relatively simple datasets to illustrate the main ideas of our work. Finally, Section 5 offers directions for future research.

## 2   Prior Work

This work was heavily inspired by the philosophy revived by Hinton et al. [5], which introduced "capsule" units. In that work, an equivariant representation is learned by the capsules when the true latent states were provided to the network as implicit targets. Our work allows us to move to a more unsupervised setting in which the true latent states are not only unknown, but represent completely arbitrary qualities. This was made possible with two assumptions: (1) that temporally adjacent samples also correspond to neighbors in the latent space, (2) predictions of future samples can be formulated as *linear* operations in the latent space. In theory, the representation learned by our method is very similar to the representation learned by the "capsules"; this representation has a locally stable *"what"* component and a locally linear, or equivariant *"where"* component. Theoretical properties of linearizing features were studied in [3].

Several recent works propose schemes for learning representations from video which use varying degrees of supervision[12][14][13][4]. For instance, [13] assumes that the pre-trained network from [7] is already available and training consists of learning to mimic this network. Similarly, [14] learns a representation by receiving supervision from a tracker. This work is more closely related to fully unsupervised approaches for learning representations from video such as [4][6][2][15][8]. It is most related to [12] which also trains a decoder to explicitly predict video frames. Our proposed architecture was inspired by those presented in in [11] and [16].

## 3   Learning Linearized Representations

Our goal is to obtain a representation of each input sequence that varies linearly in time by transforming each frame *individually*. Furthermore, we assume that this transformation can be learned by a deep, feed forward network referred to as the *encoder*, denoted by the function $F_W$. Denote the code for frame $x^t$ by $z^t = F_W(x^t)$. Assume that the dataset is parameterized by a temporal index $t$ so it is described by the sequence $X = \{..., x^{t-1}, x^t, x^{t+1}, ...\}$ with a corresponding feature sequence produced by the encoder $Z = \{..., z^{t-1}, z^t, z^{t+1}, ...\}$. Thus our goal is to train $F_W$ to produce a sequence $Z$ whose average local curvature is smaller than sequence $X$. A scale invariant local measure of curvature is the cosine distance between the two vectors formed by three temporally adjacent samples. However, minimizing the curvature directly can result in the trivial solutions: $z_t = ct \; \forall \; t$ and $z_t = c \; \forall \; t$. These solutions are trivial because they are virtually *uninformative* with respect to the input $x^t$ and therefore cannot be a *meaningful representation of the input*. To avoid this solution, we also minimize the prediction error in the input space. The predicted frame is generated in two steps: (i) linearly extrapolation in code space to obtain a predicted code $\hat{z}^{t+1} = \mathbf{a}[z^t \; z^{t-1}]^T$ followed by (ii) a *decoding* with $G_W$, which generates the predicted frame $\hat{x}^{t+1} = G_W(\hat{z}^{t+1})$. For example, if $\mathbf{a} = [2, -1]$ the predicted code $\hat{z}^{t+1}$ corresponds to a constant speed linear extrapolation of $z^t$ and $z^{t-1}$. The $L^2$ prediction error is minimized by jointly training the encoder and decoder networks. Note that minimizing prediction error alone will not necessarily lead to low curvature trajectories in $Z$ since the decoder is unconstrained; the decoder may learn a many to one mapping which maps different codes to the same output image without forcing them to be equal. To prevent this, we add an explicit curvature penalty to the loss, corresponding to the cosine distance between $(z^t - z^{t-1})$ and $(z^{t+1} - z^t)$. The complete loss to minimize is:

$$L = \frac{1}{2}\|G_W(\mathbf{a}\begin{bmatrix} z^t & z^{t-1} \end{bmatrix}^T) - x^{t+1}\|_2^2 - \lambda \frac{(z^t - z^{t-1})^T(z^{t+1} - z^t)}{\|z^t - z^{t-1}\|\|z^{t+1} - z^t\|} \tag{1}$$

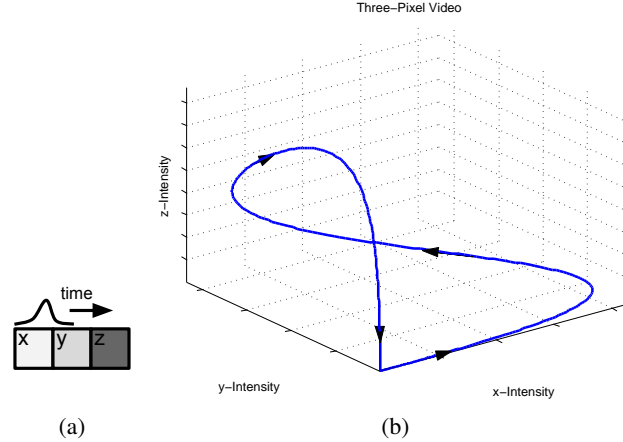

(a)               (b)

Figure 1: (a) A video generated by translating a Gaussian intensity bump over a three pixel array $(x,y,z)$, (b) the corresponding manifold parametrized by time in three dimensional space

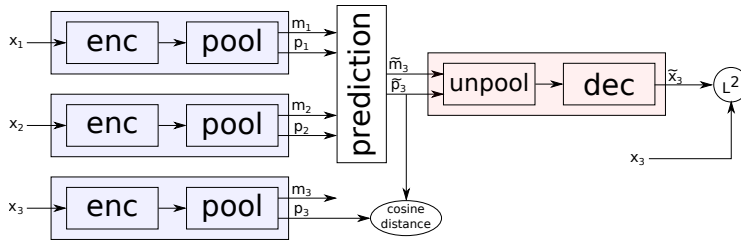

Figure 2: The basic linear prediction architecture with shared weight encoders

This feature learning scheme can be implemented using an autoencoder-like network with shared encoder weights.

## 3.1 Phase Pooling

Thus far we have assumed a generic architecture for $F_W$ and $G_W$. We now consider custom architectures and operators that are particularly suitable for the task of linearization. To motivate the definition of these operators, consider a video generated by translating a Gaussian "intensity bump" over a three pixel region at constant speed. The video corresponds to a one dimensional manifold in three dimensional space, i.e. a curve parameterized by time (see Figure 1). Next, assume that some convolutional feature detector fires only when centered on the bump. Applying the $max$-pooling operator to the activations of the detector in this three-pixel region signifies the presence of the feature somewhere in this region (*i.e. the "what"*). Applying the $argmax$ operator over the region returns the position (*i.e. the "where"*) with respect to some local coordinate frame defined over the pooling region. This position variable varies linearly as the bump translates, and thus parameterizes the curve in Figure 1b. These two channels, namely the *what* and the *where*, can also be regarded as generalized magnitude $m$ and phase $\mathbf{p}$, corresponding to a factorized representation: the magnitude represents the active set of parameters, while the phase represents the set of local coordinates in this active set. We refer to the operator that outputs both the $max$ and $argmax$ channels as the "phase-pooling" operator.

In this example, spatial pooling was used to linearize the translation of a fixed feature. More generally, the phase-pooling operator can locally linearize arbitrary transformations if pooling is performed not only spatially, but also across features in some topology.

In order to be able to back-propagate through $\mathbf{p}$, we define a soft version of the $max$ and $argmax$ operators within each pool group. For simplicity, assume that the encoder has a fully convolutional architecture which outputs a set of feature maps, possibly of a different resolution than the input. Although we can define an arbitrary topology in feature space, for now assume that we have the

familiar three-dimensional spatial feature map representation where each activation is a function $z(f, x, y)$, where $x$ and $y$ correspond to the spatial location, and $f$ is the feature map index. Assuming that the feature activations are positive, we define our soft *"max-pooling"* operator for the $k^{th}$ neighborhood $N_k$ as:

$$m_k = \sum_{N_k} z(f, x, y) \frac{e^{\beta z(f,x,y)}}{\sum_{N_k} e^{\beta z(f\prime, x\prime, y\prime)}} \approx \max_{N_k} z(f, x, y), \qquad (2)$$

where $\beta \geq 0$. Note that the fraction in the sum is a softmax operation (parametrized by $\beta$), which is positive and sums to one in each pooling region. The larger the $\beta$, the closer it is to a unimodal distribution and therefore the better $m_k$ approximates the *max* operation. On the other hand, if $\beta = 0$, Equation 2 reduces to *average-pooling*. Finally, note that $m_k$ is simply the expected value of $z$ (restricted to $N_k$) under the softmax distribution.

Assuming that the activation pattern within each neighborhood is approximately unimodal, we can define a soft versions of the $argmax$ operator. The vector $\mathbf{p}_k$ approximates the local coordinates in the feature topology at which the max activation value occurred. Assuming that pooling is done volumetrically, that is, spatially *and* across features, $\mathbf{p}_k$ will have three components. In general, the number of components in $\mathbf{p}_k$ is equal to the dimension of the topology of our feature space induced by the pooling neighborhood. The dimensionality of $\mathbf{p}_k$ can also be interpreted as the *maximal intrinsic dimension of the data*. If we define a local standard coordinate system in each pooling volume to be bounded between -1 and +1, the soft "*argmax-pooling*" operator is defined by the vector-valued sum:

$$\mathbf{p}_k = \sum_{N_k} \begin{bmatrix} f \\ x \\ y \end{bmatrix} \frac{e^{\beta z(f,x,y)}}{\sum_{N_k} e^{\beta z(f\prime, x\prime, y\prime)}} \approx \arg\max_{N_k} z(f, x, y), \qquad (3)$$

where the indices $f, x, y$ take values from -1 to 1 in equal increments over the pooling region. Again, we observe that $\mathbf{p}_k$ is simply the expected value of $\begin{bmatrix} f & x & y \end{bmatrix}^T$ under the softmax distribution.

The phase-pooling operator acts on the output of the encoder, therefore it can simply be considered as the last encoding step. Correspondingly we define an "*un-pooling*" operation as the first step of the decoder, which produces reconstructed activation maps by placing the magnitudes $m$ at appropriate locations given by the phases $\mathbf{p}$.

Because the phase-pooling operator produces both magnitude and phase signals for each of the two input frames, it remains to define the predicted magnitude and phase of the third frame. In general, this linear extrapolation operator can be learned, however "hard-coding" this operator allows us to place implicit priors on the magnitude and phase channels. The predicted magnitude and phase are defined as follows:

$$m^{t+1} = \frac{m^t + m^{t-1}}{2} \qquad (4)$$

$$\mathbf{p}^{t+1} = 2\mathbf{p}^t - \mathbf{p}^{t-1} \qquad (5)$$

Predicting the magnitude as the mean of the past imposes an implicit stability prior on $m$, i.e. the temporal sequence corresponding to the $m$ channel should be stable between adjacent frames. The linear extrapolation of the phase variable imposes an implicit linear prior on $\mathbf{p}$. Thus such an architecture produces a factorized representation composed of a locally stable $m$ and locally linearly varying $\mathbf{p}$. When phase-pooling is used curvature regularization is only applied to the $\mathbf{p}$ variables. The full prediction architecture is shown in Figure 2.

## 3.2 Addressing Uncertainty

Natural video can be inherently unpredictable; objects enter and leave the field of view, and out of plane rotations can also introduce previously invisible content. In this case, the prediction should correspond to the *most likely outcome* that can be learned by training on similar video. However, if multiple outcomes are present in the training set then minimizing the $L^2$ distance to these multiple outcomes induces the network to predict the *average outcome*. In practice, this phenomena results in blurry predictions and may lead the encoder to learn a less discriminative representation of the input. To address this inherent unpredictability we introduce latent variables $\delta$ to the prediction architecture that are not deterministic functions of the input. These variables can be adjusted *using the target*

$x^{t+1}$ in order to minimize the prediction $L^2$ error. The interpretation of these variables is that they explain all aspects of the prediction that are not captured by the encoder. For example, $\delta$ can be used to switch between multiple, equally likely predictions. It is important to control the capacity of $\delta$ to prevent it from explaining the entire prediction on its own. Therefore $\delta$ is restricted to act only as a correction term in the code space output by the encoder. To further restrict the capacity of $\delta$ we enforce that $dim(\delta) \ll dim(z)$. More specifically, the $\delta$-corrected code is defined as:

$$\hat{z}_{\delta}^{t+1} = z^t + (W_1\delta) \odot \mathbf{a} \begin{bmatrix} z^t & z^{t-1} \end{bmatrix}^T \tag{6}$$

Where $W_1$ is a trainable matrix of size $dim(\delta) \times dim(z)$, and $\odot$ denotes the component-wise product. During training, $\delta$ is inferred (using gradient descent) for each training sample by minimizing the loss in Equation 7. The corresponding adjusted $\hat{z}_{\delta}^{t+1}$ is then used for back-propagation through $W$ and $W_1$. At test time $\delta$ can be selected via sampling, assuming its distribution on the training set has been previously estimated.

$$L = \min_{\delta} \|G_W(\hat{z}_{\delta}^{t+1}) - x^{t+1}\|_2^2 - \lambda \frac{(z^t - z^{t-1})^T(z^{t+1} - z^t)}{\|z^t - z^{t-1}\|\|z^{t+1} - z^t\|} \tag{7}$$

The following algorithm details how the above loss is minimized using stochastic gradient descent:

---

**Algorithm 1** Minibatch stochastic gradient descent training for prediction with uncertainty. The number of $\delta$-gradient descent steps ($k$) is treated as a hyper-parameter.

---

    **for** number of training epochs **do**
        Sample a mini-batch of temporal triplets $\{x^{t-1}, x^t, x^{t+1}\}$
        Set $\delta_0 = 0$
        Forward propagate $x^{t-1}, x^t$ through the network and obtain the codes $z^{t-1}, z^t$ and the prediction $\hat{x}_0^{t+1}$
        **for** $i = 1$ to $k$ **do**
            Compute the $L^2$ prediction error
            Back propagate the error through the decoder to compute the gradient $\frac{\partial L}{\partial \delta^{i-1}}$
            Update $\delta_i = \delta_{i-1} - \alpha \frac{\partial L}{\partial \delta^{i-1}}$
            Compute $\hat{z}_{\delta_i}^{t+1} = z^t + (W_1\delta_i) \odot \mathbf{a} \begin{bmatrix} z^t & z^{t-1} \end{bmatrix}^T$
            Compute $\hat{x}_i^{t+1} = G_W(z_{\delta^i}^{t+1})$
        **end for**
        Back propagate the full encoder/predictor loss from Equation 7 using $\delta_k$, and update the weight matrices $W$ and $W_1$
    **end for**

---

When phase pooling is used we allow $\delta$ to only affect the phase variables in Equation 5, this further encourages the magnitude to be stable and places all the uncertainty in the phase.

## 4 Experiments

The following experiments evaluate the proposed feature learning architecture and loss. In the first set of experiments we train a shallow architecture on natural data and visualize the learned features in order gain a basic intuition. In the second set of experiments we train a deep architecture on simulated movies generated from the NORB dataset. By generating frames from interpolated and extrapolated points in code space we show that a linearized representation of the input is learned. Finally, we explore the role of uncertainty by training on only partially predictable sequences, we show that our latent variable formulation can account for this uncertainty enabling the encoder to learn a linearized representation even in this setting.

### 4.1 Shallow Architecture Trained on Natural Data

To gain an intuition for the features learned by a phase-pooling architecture let us consider an encoder architecture comprised of the following stages: convolutional filter bank, rectifying point-wise nonlinearity, and phase-pooling. The decoder architecture is comprised of an un-pooling stage followed by a convolutional filter bank. This architecture was trained on simulated $32 \times 32$ movie

|  | Encoder | Prediction | Decoder |
|---|---|---|---|
| Shallow Architecture 1 | Conv+ReLU 64 $\times$ 9 $\times$ 9<br>Phase Pool 4 | Average Mag.<br>Linear Extrap. Phase | Conv 64 $\times$ 9 $\times$ 9 |
| Shallow Architecture 2 | Conv+ReLU 64 $\times$ 9 $\times$ 9<br>Phase Pool 4 stride 2 | Average Mag.<br>Linear Extrap. Phase | Conv 64 $\times$ 9 $\times$ 9 |
| Deep Architecture 1 | Conv+ReLU 16 $\times$ 9 $\times$ 9<br>Conv+ReLU 32 $\times$ 9 $\times$ 9<br>FC+ReLU 8192 $\times$ 4096 | None | FC+ReLU **8192** $\times$ 8192<br>Reshape 32 $\times$ 16 $\times$ 16<br>SpatialPadding 8 $\times$ 8<br>Conv+ReLU 16 $\times$ 9 $\times$ 9<br>SpatialPadding 8 $\times$ 8<br>Conv 1 $\times$ 9 $\times$ 9 |
| Deep Architecture 2 | Conv+ReLU 16 $\times$ 9 $\times$ 9<br>Conv+ReLU 32 $\times$ 9 $\times$ 9<br>FC+ReLU 8192 $\times$ 4096 | Linear Extrapolation | FC+ReLU 4096 $\times$ 8192<br>Reshape 32 $\times$ 16 $\times$ 16<br>SpatialPadding 8 $\times$ 8<br>Conv+ReLU 16 $\times$ 9 $\times$ 9<br>SpatialPadding 8 $\times$ 8<br>Conv 1 $\times$ 9 $\times$ 9 |
| Deep Architecture 3 | Conv+ReLU 16 $\times$ 9 $\times$ 9<br>Conv+ReLU 32 $\times$ 9 $\times$ 9<br>FC+ReLU 8192 $\times$ 4096<br>Reshape 64 $\times$ 8 $\times$ 8<br>Phase Pool 8 $\times$ 8 | Average Mag.<br>Linear Extrap. Phase | Unpool 8 $\times$ 8<br>FC+ReLU 4096 $\times$ 8192<br>Reshape 32 $\times$ 16 $\times$ 16<br>SpatialPadding 8 $\times$ 8<br>Conv+ReLU 16 $\times$ 9 $\times$ 9<br>SpatialPadding 8 $\times$ 8<br>Conv 1 $\times$ 9 $\times$ 9 |

Table 1: Summary of architectures

frames taken from YouTube videos [4]. Each frame triplet is generated by transforming still frames with a sequence of three rigid transformations (translation, scale, rotation). More specifically let $A$ be a random rigid transformation parameterized by $\tau$, and let $x$ denote a still image reshaped into a column vector, the generated triplet of frames is given by $\{f_1 = A_{\tau=\frac{1}{3}}x, f_2 = A_{\tau=\frac{2}{3}}x, f_3 = A_{\tau=1}x\}$. Two variants of this architecture were trained, their full architecture is summarized in the first two lines of Table 1. In Shallow Architecture 1, phase pooling is performed spatially in non-overlapping groups of $4 \times 4$ and across features in a one-dimensional topology consisting of non-overlapping groups of four. Each of the 16 pool-groups produce a code consisting of a scalar $m$ and a three-component $\mathbf{p} = [p_f, p_x, p_y]^T$ (corresponding to two spatial and one feature dimensions); thus the encoder architecture produces a code of size $16 \times 4 \times 8 \times 8$ for each frame. The corresponding filters whose activations were pooled together are laid out horizontally in groups of four in Figure 3(a). Note that each group learns to exhibit a strong ordering corresponding to the linearized variable $p_f$. Because global rigid transformations can be locally well approximated by translations, the features learn to parameterize local translations. In effect the network learns to linearize the input by tracking common features in the video sequence. Unlike the spatial phase variables, $p_f$ can linearize sub-pixel translations. Next, the architecture described in column 2 of Table 1 was trained on natural movie patches with the natural motion present in the real videos. The architecture differs in only in that pooling across features is done with overlap (groups of 4, stride of 2). The resulting decoder filters are displayed in Figure 3 (b). Note that pooling with overlap introduces smoother transitions between the pool groups. Although some groups still capture translations, more complex transformations are learned from natural movies.

## 4.2 Deep Architecture trained on NORB

In the next set of experiments we trained deep feature hierarchies that have the capacity to linearize a richer class of transformations. To evaluate the properties of the learned features in a controlled setting, the networks were trained on simulated videos generated using the NORB dataset rescaled to $32 \times 32$ to reduce training time. The simulated videos are generated by tracing constant speed trajectories with random starting points in the two-dimensional latent space of pitch and azimuth rotations. In other words, the models are trained on triplets of frames ordered by their rotation angles. As before, presented with two frames as input, the models are trained to predict the third frame. Recall that *prediction is merely a proxy for learning linearized feature representations*. One way to evaluate the linearization properties of the learned features is to linearly interpolate (or extrapolate)

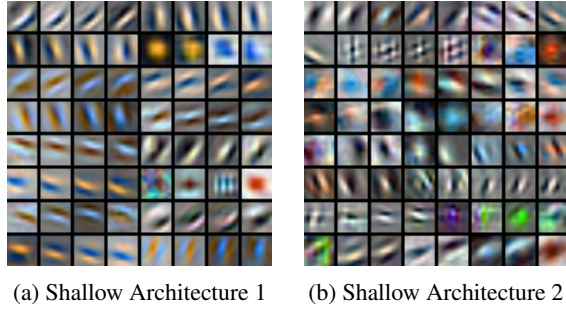

(a) Shallow Architecture 1          (b) Shallow Architecture 2

Figure 3: Decoder filters learned by shallow phase-pooling architectures

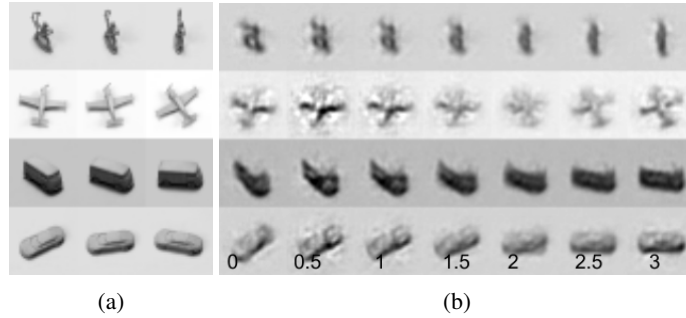

(a)                                        (b)

Figure 4: (a) Test samples input to the network (b) Linear interpolation in code space learned by our Siamese-encoder network

new codes and visualize the corresponding images via forward propagation through the decoder. This simultaneously tests the encoder's capability to linearize the input and the decoder's (generative) capability to synthesize images from the linearized codes. In order to perform these tests we must have an *explicit* code representation, which is not always available. For instance, consider a simple scheme in which a generic deep network is trained to predict the third frame from the concatenated input of two previous frames. Such a network does not even provide an explicit feature representation for evaluation. A simple baseline architecture that affords this type of evaluation is a Siamese encoder followed by a decoder, this exactly corresponds to our proposed architecture with the linear prediction layer removed. Such an architecture is equivalent to *learning the weights of the linear prediction layer* of the model shown in Figure 2. In the following experiment we evaluate the effects of: (1) fixing v.s. learning the linear prediction operator, (2) including the phase pooling operation, (3) including explicit curvature regularization (second term in Equation 1).

Let us first consider Deep Architecture 1 summarized in Table 1. In this architecture a Siamese encoder produces a code of size 4096 for each frame. The codes corresponding to the two frames are concatenated together and propagated to the decoder. In this architecture the first linear layer of the decoder can be interpreted as a learned linear prediction layer. Figure 4a shows three frames from the test set corresponding to temporal indices 1,2, and 3, respectively. Figure 4b shows the generated frames corresponding to interpolated codes at temporal indices: $0, 0.5, 1, 1.5, 2, 2.5, 3$. The images were generated by propagating the corresponding codes through the decoder. Codes corresponding to non-integer temporal indices were obtained by linearly interpolating in code space.

Deep Architecture 2 differs from Deep Architecture 1 in that it generates the predicted code via a fixed linear extrapolation in code space. The extrapolated code is then fed to the decoder that generates the predicted image. Note that the fully connected stage of the decoder has half as many free parameters compared to the previous architecture. This architecture is further restricted by propagating only the predicted code to the decoder. For instance, unlike in Deep Architecture 1, the decoder cannot copy any of the input frames to the output. The generated images corresponding to this architecture are shown in Figure 5a. These images more closely resemble images from the dataset. Furthermore, Deep Architecture 2 achieves a lower $L^2$ prediction error than Deep Architecture 1.

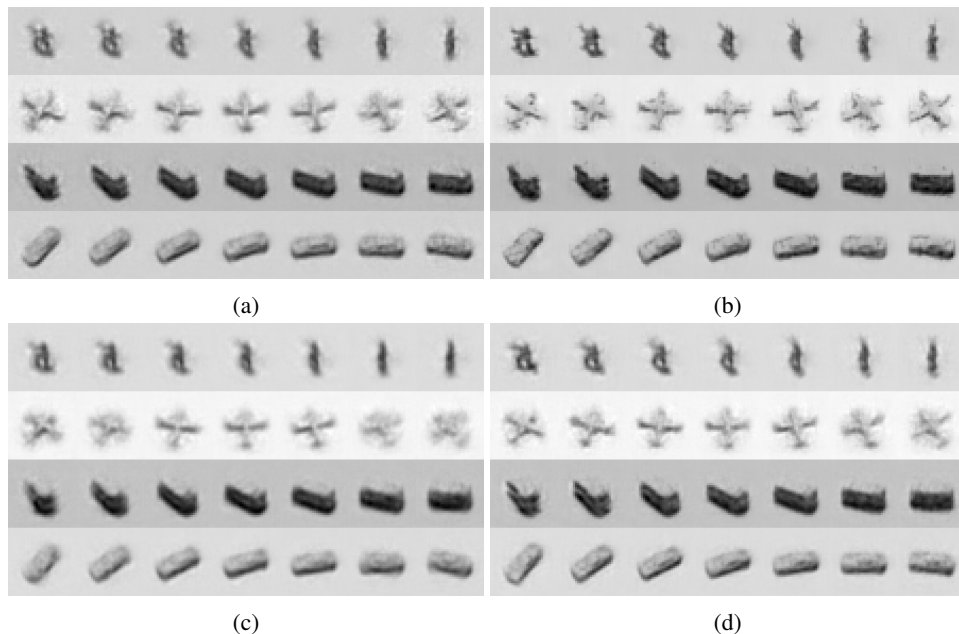

Figure 5: Linear interpolation in code space learned by our model. (a) no phase-pooling, no curvature regularization, (b) with phase pooling and curvature regularization Interpolation results obtained by minimizing (c) Equation 1 and (d) Equation 7 trained with only partially predictable simulated video

Finally, Deep Architecture 3 uses phase-pooling in the encoder, and "un-pooling" in the decoder. This architecture makes use of phase-pooling in a two-dimensional feature space arranged on an $8 \times 8$ grid. The pooling is done in a single group over all the fully-connected features producing a feature vector of dimension $192$ $(64 \times 3)$ compared to $4096$ in previous architectures. Nevertheless this architecture achieves the best overall $L^2$ prediction error and generates the most visually realistic images (Figure 5b). In this subsection we compare the representation learned by minimizing the loss in Equation 1 to Equation 7. Uncertainty is simulated by generating triplet sequences where the third frame is skipped randomly with equal probability, determined by Bernoulli variable $s$. For example, the sequences corresponding to models with rotation angles $0°, 20°, 40°$ and $0°, 20°, 60°$ are equally likely. Minimizing Equation 1 with Deep Architecture 3 results in the images displayed in Figure 5c. The interpolations are blurred due to the averaging effect discussed in Subsection 3.2. On the other hand minimizing Equation 7 (Figure 5d) partially recovers the sharpness of Figure 5b. For this experiment, we used a three-dimensional, real valued $\delta$. Moreover training a linear predictor to infer binary variable $s$ from $\delta$ (after training) results in a $94\%$ test set accuracy. This suggests that $\delta$ does indeed capture the uncertainty in the data.

## 5   Discussion

In this work we have proposed a new loss and architecture for learning locally linearized features from video. We have also proposed a method that introduces latent variables that are non-deterministic functions of the input for coping with inherent uncertainty in video. In future work we will suggest methods for "stacking" these architectures that will linearize more complex features over longer temporal scales.

**Acknowledgments**

We thank Jonathan Tompson, Joan Bruna, and David Eigen for many insightful discussions. We also gratefully acknowledge NVIDIA Corporation for the donation of a Tesla K40 GPU used for this research.

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
