[Reviews · NeurIPS 2015]

Submitted by Assigned_Reviewer_1

Summary:

The paper addresses two main issues with the current autoencoder representations for the video frame prediction task:

-- Using L2 loss to minimize the difference between the predicted and reference frame leads to blurry representations, as the network averages under uncertainty. A latent variable formulation is introduced to address this, where separate variables that are not coupled deterministically with the frame inputs are allowed to affect the output.

-- It is generally difficult to learn expressive representations, as the identify function from the most recent frame is a strong predictor of the result. The paper proposes decoupling the representation into a locally stable "what" component (the weighted average of the final feature map) and a locally linear "where" component (the phase, or soft argmax of the final feature map activations). A loss function enforcing that the phase of the predicted frame can be linearly extrapolated from the phases of the previous two frames.

The experimental results demonstrate mostly anecdotally that representations with the desired properties (what are where components, and latent variables) are indeed learned and positively impact video frame reconstruction quality.

Quality:

I did not see notable issues in the intuitions or technical implementation. The arguments are substantiated with intuitive results on a toy experimental dataset (NORB). Some questions:

-- Is the beta parameter from Eqns 2 and 3 learned. If not (as it is not present in Algorithm 1 pseudocode), then how was it set and what was its effect?

-- The paper mentions that "at test time  can be selected via sampling, assuming its distribution on the training set has been previously estimated". It would be helpful to visualize how sampling delta affects the results. It would not only help us visualize the uncertainty that is being learned, but also verify the hypothesis above whether sampling is a viable options of getting believable-looking frames.

-- Deep architecture 3 has a bit of an unusual design. What is the intuition behind computing a fully-connected layer only to reshape it back into a convolutional layer? Does a fully-convolutional design work and if not, why not?

-- The exposition mentions the L2 reconstruction errors of different deep architectures, but I did not see a table with the values anywhere (or how they compare to the other frame-reconstruction methods). It would be helpful to have numeric estimates, not just visual-anecdotal ones.

Clarity:

The paper is clearly written, except a couple of spots. I had trouble parsing Table1 -- while succinct, without knowing the filter sizes and phase pooling sizes it is hard to intuit what happens in the architectures. I also had trouble following some of the exact details of Sec 4.1 for the same reason -- but the deep network experiment explanations were fine.

Originality:

The paper has several original ideas: 1) phase pooling, including its soft-argmax computation and the associated phase linear extrapolation loss 2) the introduction of the latent variables delta. To the best of my knowledge, these are original ideas and a creative way of using autoencoder models for video frame prediction.

Significance:

The paper expands the kinds of auto-encoder representations that can be learned from video and provide intriguing enough results that I believe may spur interesting follow-up work.

Summary: The paper introduces several different ideas for learning video autoencoder-based representations. It decouples the representation into 3 separate parts (the "what", the "where" and the "prediction uncertainty") and introduces a novel loss and a setup to train it. The paper has sufficient novelty and significance for publication. The experimental section succeeds in illustrates the expected representation properties, although numerical evaluation and more comparisons to related work would be of additional help.

Submitted by Assigned_Reviewer_2

This paper proposes a new loss and architecture for learning linearized features from unlabeled video frames. It also introduces latent variables, which are non-deterministic functions of the input, into the architecture to solve the problems caused by inherent uncertainty in video.

The paper aims to train deep feature hierarchies in a weak-supervision manner, which is an emerging research direction. one

novel idea is to learn linearized features. An explicit code representation for every single frame is given, which

makes it easy to linearly interpolate or extrapolate new codes for new frames. The idea to cope with the inherent uncertainty in video is also interesting and worth trying.

The presentation is rather techinical, requiring the readers to be familiar with details in this area. There are several problems with this paper. Firstly, the title might be too broad, as the proposed method is designed for video data, and cannot deal with, for instance, text data. Secondly, the experiment part seems insufficient for several reasons. The frames are all of small size (32*32) and the data set for the second experiment is also small and toy. Although the authors claim to learn features from natural video, the frame triplet for the experiments are mostly simulated videos. Natural movie patches are used in one setting of experiment in the paper, but it only shows us the learned filters as results, which is not enough for us to see the effectiveness of the method on natural videos. The exact L2 prediction errors for the several experiment settings in 4.2 are also missing. We can only compare the results by the generated frames without quantitative analysis. Moreover, it would be good if the authors could provide us with results for translation and scale appeared in video frames, and can spend more efforts on explaining the architectures in table1.

Questions:

1) What is the exact setting for the value of a in equations (1) and (6) in the experiments?

2) For (6), is there still a linear relationship between zt-1, zt and zt+1? If not, why is it said to be learning linearized features?
Summary: The paper proposes novel ideas to cope with the inherent uncertainty in video and learn linearized features from video frames. However, the experiment part seems insufficient for a NIPS paper.

Submitted by Assigned_Reviewer_3

The authors present a method to linearize video features in latent space while accounting for uncertainty in the features as a function of frame progression. There are some good ideas in this paper, but they feel unready and are not supported by by convincing experiments demonstrating the linearization and the precise role of the latent variable delta in test time.
Summary: The authors present a deep learning architecture tailored towards video and aim to linearize the complex transformations involved in video processing. While the ideas are good and preliminary experiments are shown, the model seems unready and is not convincing as a linearization of video content.

Submitted by Assigned_Reviewer_4

This paper introduces a new loss function and a novel deep learning architecture to learn locally linearized features from video. This paper is well written and organized. The problem studied in this work, i.e., learning locally linearized features from video in order to predict video frames, is an important and interesting problem to study. The proposed architecture and objective in Eq. (7) seems novel and very interesting.

The main concern is about the empirical evaluation.

Unlike the other works, e.g., [8][12][13][14], the authors only train and evaluate the proposed approach over simple toy video sequences. For real world applications, natural video sequences should be used to demonstrate the power of the proposed approach.

The current results can only be evaluated subjectively, i.e., different judgers may has different opinions for whether a prediction result is correct or incorrect (correct to what degree or incorrect to what degree). Therefore, it would be nice to provide a quantitative measure for all the output.

Several related works were mentioned (e.g., [8][12][13][14]), but none of them was compared to demonstrate the effectiveness of the proposed approach. This makes the justification of the proposed approach unconvincing.
Summary: This paper introduces a new loss and architecture to learn locally linearized features from video.

The main idea is to train a generative model to predict video frames. The experiment shows the effectiveness of their proposed approach.

Submitted by Assigned_Reviewer_5

Section 3.1 seems unnecessarily hard to understand. The models could be better described by fixing Figure 2: 1) making the notation consistent (x_1 vs x_t), 2) putting z_t, z_{t-1}, z_{t+1}, F_W, G_W explicitly in the picture, 3) in the text, matching the cosine label in Figure 2, to the math of (1) via z.

Section 3.2 and 4.3 didn't make sense to me. Indexing examples in this way does not conform to any notion of uncertainty I familiar with (but would be happy to be enlightened with references!)
Summary: The authors propose a autoencoder architecture on triples that attempts to linearly interpolate in some latent space to find smooth trajectories. I am rather confused by the "uncertainty" in the paper, as there does not appear to be any (beyond any notion of uncertainty one might argue for in any autoencoder). Delta appears to be an interpolation parameter. Other than that, the paper is interesting, and the results are neat.

Author Feedback
Author rebuttal: We sincerely thank all the reviewers for their thoughtful reviews.

Reviewer-1
1-Thank you for pointing out this relevant related work by Desjardins et al. Similar to this work, our experiments on NORB show that linearization can implicitly disentangle factors of variation (rotation angles).
2-We agree that applying our method to action recognition datasets would be a good test in future works.

Reviewer-2
1-Our work is presented in the most elementary way we found possible.
2-The title is intentionally broad because our method can be applied to any temporally coherent data.
3-Indeed it is difficult to quantify the performance of learned features in the unsupervised setting. However, our experiments demonstrate that the learned features linearize the latent factors of variation in NORB. Similar evaluations were carried out recently in [3]. See comment 1 to Reviewer 6.
4-We did not report the L2 prediction error because it is not always commensurate with the visual quality of the predicted image, even more so if the temporal sequence is not fully deterministic. Nevertheless we can report a normalized L2 prediction error in the final version.
5-For an explanation of the architecture, please see comment 4 to Reviewer-4
6-We will elaborate on the experimental details with natural and simulated video in the final version.
7(Q1)-a = [2 -1] in Equations 1&6
8(Q2)-The \delta variable can be seen as a slack variable in code space, allowing the predicted code to be shifted slightly in order to generate the ground truth. However we neglected to say that not only the dimensionality, but also the norm of \delta is constrained, enforcing a nearly linearized code.

Reviewer-3
1-See comment 1 to Reviewer-6.
2-The mentioned works often rely on much more supervision than our method. For example, [8] trains in a semi-supervised setting on the COIL dataset, which like NORB, also consists of sequences of rotating objects. Although [13] and [12] are trained on natural video sequences, they are both trained with supervision, for example [13] is only trained to mimic the activations of a network trained with supervision. [14] is the only generative model in this list that is trained in an unsupervised manner, however it also relies on mainly qualitative evaluation.
3-Please see comments 3&4 to Reviewer-2.

Reviewer-4
1-See comments 3&4 to Reviewer-2.
2-We did not try learning \beta in our implementation. We chose it manually based on the visual quality of the predicted images. We found that it was necessary to chose \beta large enough (>5) to produce approximately unimodal distributions of activations within each pooling region. We also found that once \beta was large enough, our method was quite robust to its exact value.
3-We can visualize predictions corresponding to different \delta samples in the camera-ready. Although these predictions do not necessarily correspond to the lowest L2 prediction error, they are more visually plausible. Therefore, the \delta formulation implicitly modifies the L2 metric to account for uncertainty.
4-Our intuition for including a fully connected layer in our architecture follows from the fact that the latent variables in the NORB dataset have no spatial structure. That is, the rotation angles are global parameters that correspond to the entire image. Fully convolutional architectures performed worse in our experiments.
5-We can include the L2 prediction errors, however their relevance is questionable. See comment #4 to Reviewer-2.
6-We will rephrase some parts of section 4.1 to make them more clear.

Reviewer-5
1-In sec 4.3 we manually introduce uncertainty in temporal speed by skipping frames randomly with equal probability. The interpretation is that \delta switches between these two possibilities, which implicitly modifies the L2 loss to account for uncertainty. See comment 8 to Reviewer-2.
2-The notation in Figure 2 will be made consistent with the paper.

Reviewer-6
1-The objective of this paper is to introduce temporal linearization as a principled prior for unsupervised feature learning. It also proposes novel operators for achieving this goal, and addresses the problem of uncertainty in prediction. Promising results are presented on controlled dataset where the true latent factors of variations are known. This would be more difficult to demonstrate in natural unlabeled video because the true factors are unknown. Similar evaluations were used in [3], Hadsell et al CVPR 2006, and many other related works. This paper presents the building blocks towards achieving the ultimate goal of learning useful features from natural unlabeled videos. We believe that a complete solution to this broad and difficult problem will require a much larger effort that warrants many other follow up publications.
2-See comment 3 to Reviewer-4.